# Challenges in Help-Seeking Behaviors among Rural Older People Mitigated through Family Physician-Driven Outreach: A Systematic Review

**DOI:** 10.3390/ijerph192417004

**Published:** 2022-12-18

**Authors:** Ryuichi Ohta, Takuji Katsube, Chiaki Sano

**Affiliations:** 1Community Care, Unnan City Hospital, Unnan 699-1221, Japan; 2Department of Community Medicine Management, Faculty of Medicine, Shimane University, Izumo 690-0823, Japan

**Keywords:** help-seeking, rural, family medicine, physician, outreach, older people, elderly

## Abstract

Help-seeking behaviors (HSBs) refer to approaches taken by individuals towards their health and symptoms, and they are supported by healthcare professionals. Outreach interventions aimed at older people in rural communities can mitigate difficulties in implementing HSBs and help them remain healthy. This systematic review investigated evidence regarding family medicine-involved outreach aimed at HSBs among older individuals in rural areas. We searched three databases (PubMed, EMBASE, and Web of Science) for international and original interventional articles regarding family physicians involved in outreach to older people in rural or underserved areas between April 2000 and October 2022. The articles were analyzed and summarized based on the setting, country, health issues, and outreach outcomes. Of the 376 studies identified, four were included in this review. Our findings showed that family physician-involved outreach to rural and underserved areas improved health outcomes, including anxiety, subjective physical function, and diabetic care. The challenges of outreach interventions include the duration and continuity of outreach, the active participation of family physicians and patients in the outreach programs, and the focus of outreach participants. Although the number of studies included was small, family physician-involved outreach to rural and underserved areas was shown to improve various health outcomes.

## 1. Introduction

An individual’s health can be affected by their approaches to health and symptoms in their everyday lives. These health-related behaviors are known as help-seeking behaviors (HSBs), which refer to concrete behaviors, including taking rest, gathering information, and consulting with relatives and healthcare professionals [1,2]. HSBs are categorized into lay and professional care. Lay care is provided by those with lay knowledge and non-professionals [1]. It involves self-management; gathering knowledge; consulting with families, relatives, and friends; buying and using over-the-counter drugs; and home remedies [1,2]. Meanwhile, professional care is provided by professionals and involves visiting primary care doctors, pharmacists, and emergency rooms in general hospitals [1,2]. According to their symptoms, effective lay and professional care are critical for people’s health conditions [3,4].

HSBs may be related to subjective health conditions, including quality of life (QOL). Previous research has demonstrated that self-management as a form of lay care can be related to a high QOL [5]. Another study showed that self-medication could also be associated with high QOL [6]. Furthermore, self-management of common symptoms have been shown to improve QOL, including during the 2019 coronavirus pandemic [7]. Therefore, improvements to self-management methods and medication use may improve QOL. Notably, older people tend to experience more symptoms than younger generations and use various HSBs [8,9,10]. Thus, the HSBs of older people should be modified to improve their health conditions [10]. 

HSBs may be influenced by the environment, particularly in older people. Globally, HSBs in older people are a critical public health issue [11]. Aging causes the deterioration of physical and cognitive abilities, and older people lose accessibility to various social resources, owing to the loss of capacity to drive and difficulty in using public transportation [12,13]. Moreover, living in rural areas can affect older people’s lives because of the scarcity of social resources and public transportation systems [12,13], and such conditions may prevent them from utilizing healthcare systems. Furthermore, delays in using healthcare resources may cause the progression of critical diseases, leading to morbidity and mortality. 

Therefore, in the rural context, older people’s HSBs should be improved for their health and the sustainability of rural healthcare systems. For sustainability, outreach by healthcare professionals to rural older people who cannot access healthcare institutions because of low accessibility and availability is an effective approach for early detection of modifiable risk factors using healthcare resources [14]. Among older people, delays in using healthcare resources in critical situations, including cardiovascular diseases and malignancy, are detrimental to their lives [15,16]. Therefore, outreach aimed at these populations in rural communities can mitigate the risks of acute diseases, helping them remain healthy. In addition, because aging is progressing, effective outreach to rural communities can reduce multimorbidity issues, healthcare usage irregularity, and the burden on rural healthcare professionals.

Family physicians who specialize in person-centered care and promoting health conditions should lead effective outreach projects as family medicine can address various health issues that occur within communities [17]. Regarding health promotion through HSBs, family physicians can collaborate with people and other healthcare professionals in communities to improve their perceptions of HSBs and concrete behaviors regarding their health [17]. Globally, there are various healthcare resources and professionals to help improve HSBs [12]. Clarifying evidence-based outreach interventions involving family physicians for improving HSBs can enhance family physician-driven outreach regarding HSBs. Therefore, this systematic review aimed to investigate current evidence regarding outreach involving family medicine aimed at HSBs in older people in rural areas.

## 2. Materials and Methods

This systematic review was conducted according to the PRISMA guidelines [18]. This study was registered on the PROSPERO platform with registration number 371095. In addition, we searched for interventions for HSBs related to family medicine in PubMed, Web of Science, and Embase between April 2000 and October 2022. The words used in the search were [“rural” or “remote” or “underserved”] AND [“older’’ or “elderly”] AND [“family physician” or “general practitioner” or “primary care”] AND “outreach” AND “community.” 

### 2.1. Study Selection

The inclusion and exclusion criteria are presented in Table 1. Original interventional articles were included in the international context, whereas conference presentations, reviews, and duplicate articles in the search results were excluded. 

### 2.2. Data Extraction

The literature search, data extraction, and review were conducted by three investigators (RO, TK, and CS), and any discrepancies were resolved through discussion. The databases were searched for original studies on the health promotion through HSBs. Studies without clear descriptions of the aims, participants, or outcomes were excluded (Table 1).

Concretely, one of the investigators (R.O.) extracted the data from each original study using a purpose-designed data-extraction form. Two other investigators (T.K. and C.S.) examined the extracted data, which were categorized as follows: country, publication year, participants, purpose, research methodology, health issues, types of intervention, involved professionals, and outcomes concerning outreach.

### 2.3. Statistical Analysis

This study excluded statistical analysis because of the small number of included articles. However, the data from each study are presented descriptively. The quality of each study was assessed based on the best evidence medical education scale (1 to 5): grade 1 indicated that no definite conclusions could be drawn, that is, the data were not significant; grade 2 showed that the results were ambiguous, although there appeared to be a trend; grade 3 indicated that conclusions could be drawn based on the results; grade 4 indicated that the results were clear and probably very true; grade 5 indicated that the results were unequivocal [19].

## 3. Results

Overall, 376 studies were identified. Of these, 25 duplicate studies were excluded. After reviewing the abstracts, 333 studies were excluded for the following reasons: 69, different settings; 123, different participants; 103, no interventions; and 38, no clear health outcomes. Finally, a total of four studies were identified in the final analysis after excluding 14 articles through the assessment of eligibility (10, unoriginal articles; 4, no outreach to communities) (Figure 1). The details of the four articles are presented in Table 2. Each article was summarized in the categories of study design, participants, countries, health issues, interventions, involved professionals, and outcomes.

### 3.1. Summary of the Study Results

In terms of the study designs, all the studies were comparative interventional studies, and two were randomized controlled trials. The participants were over 60–65 years of age. Two studies were from the United States [22,23] and two were from Canada [20,21]. In regard to the range of health issues, one study dealt with functional decline in usual life [21], one with worry and generalized anxiety disorder [22], one with diabetes control [20], and one with physical dysfunction [23]. The setting of two studies included rural communities [20,21], while the other two studies included underserved areas [22,23]. All of the study interventions included multiple professionals: All studies, with family physicians or general practitioners; three, with community workers; one each, with family, patients, social workers, therapists, specialists, dieticians, and pharmacists; and two, with nurses. Considering the outcome measurements, one study measured the QOL based on a questionnaire [22], one measured the worry and general anxiety disorder (GAD) severity with multiple questionnaires [21], one measured the satisfaction of care in diabetes [20], and one measured the perceived and objective physical functions using multiple questionnaires [23]. All of the studies demonstrated unequivocal data considering community outreach; the study grade was rated five.

### 3.2. Suggested limitations of the interventions

First, the involvement of family physicians in the interdisciplinary teams was limited, considering the implication on care decision-making. A previous study suggested that mutual collaboration among specialists, family physicians, and patients in chronic care can improve the objective health outcomes [20]. Second, the established primary care sufficiently supported older people, and the outreach projects may not improve the health outcomes; thus, outreach projects should be conducted in rural and underserved areas [21]. Third, the involvement of various professionals was limited. A previous study suggested that the involvement of general physicians and other healthcare professionals could improve the health outcomes of the participants [22]. Fourth, the duration of the intervention was short. Continual involvement of family physicians and multiple healthcare professionals in the healthcare of older individuals is needed for objective improvement [23].

## 4. Discussion

This study shows that family physician-involved outreach to rural and underserved areas can improve various health outcomes, including anxiety, subjective physical function, and diabetic care. The issues of outreach interventions are the duration and continuity of outreach, the active participation of family physicians and patients in outreach programs, and the focus of outreach participants.

For effective outreach, the focus should be specifically targeted to assessing rural community conditions. One article included in this systematic review showed no effectiveness of QOL changes among older people, based on the assessment of home care nurses [21]. As QOL is a patient-reported outcome, the patient’s perception is essential for improvement [24]. If the participants were satisfied with the present conditions, the additional interventions, without considering the true needs of the participants, may not change their perceptions of their QOL, eventually leading to no improvement in research outcomes [9,25]. However, other studies of outreach focusing on rural community needs regarding the gap between primary care clinics and local people improved the participants’ perceived physical functions and worries [20,22,23]. Therefore, considering the principle of family medicine, family physicians should focus on the needs of each community to improve their health conditions [26,27]. The need assessment for interventions is essential in public health for establishing effective outreach [28], and thus should include actual need assessments in communities.

The duration of outreach interventions is crucial to improve the subjective and objective outcomes in rural communities. This review reveals that some outreach involving various healthcare professionals and patients improves the subjective health conditions, including worry, subjective physical functions, and diabetic care quality [20,22,23]. However, these studies cannot change the objective health outcomes, including physical function and other chronic disease outcomes. These results could be attributed to the short intervention duration of outreach influencing changes in these outcomes. These studies may not be able to take advantage of the continuity of the care, which is crucial for improving patient health in family medicine [29,30]. Furthermore, the continuity of care improves various health outcomes in primary care [31,32]. All the included studies had less than a year of intervention. However, this duration was longer than that in other scientific studies regarding medicine. Therefore, community outreach programs should evaluate social needs and the acceptance of interventions in rural communities for effective implementation [28]. Moreover, effective acceptance of rural older people and the continuous implementation of outreach in rural communities can facilitate efficient implementation, changing the objective outcomes in healthcare [28]. Therefore, family physicians should respect the continuity of care in rural outreach programs and continue outreach to communities in collaboration with various healthcare professionals and stakeholders in order to change the objective outcomes.

The active participation of family physicians and patients in outreach programs should be promoted. Person-centered care and continuity of care are the competencies of family physicians ana are essential for effective outreach in rural contexts [26,33]. In this study, family physicians were members of outreach interdisciplinary teams [20,21,22,23]. However, the primary interventions were conducted by individuals in other specialties and healthcare professionals, including geriatricians, home care nurses, care workers, and counselors [20,21,22,23]. Each professional could approach a specific set of older individuals and patients; however, the systemic and holistic approaches in collaboration with patients, community members, and family physicians are lacking in the studies of this review. In addition, the empowerment of rural people and patients is crucial for health promotion in rural communities [34]. Without this empowerment, outreach interventions may not make sufficient changes to rural community health [35]. The reviewed studies assessed subjective and objective health rather than the perception and motivation for behavioral changes and perceptions regarding outreach. For effective outreach intervention, family physicians can create rural outreach programs in collaboration with various professionals and lay people, respecting the principles of person-centered care and the continuity of care in rural contexts [36].

This study had some limitations. First, few original studies investigated outreach programs involving family physicians and general practitioners. Many family physicians may approach this field in rural places; however, their effectiveness in rural people’s health conditions may not be clarified. Therefore, future studies should use longitudinal research designs to assess outreach programs in rural contexts regarding improving health programs, which could motivate more family physicians to conduct outreach programs in their communities. Second, this systematic review excluded articles other than interventional studies to clarify the current evidence regarding rural outreach to communities. This inclusion criterion may exclude some grey studies from rural outreach by family physicians, including the term “regional” as a search term. Third, because of accessibility limitations, the review may have missed studies published in languages other than English. However, to overcome this limitation, we used search engines worldwide. As the world population is gradually aging, all countries are experiencing the issues of aging societies, in turn necessitating rural outreach from family physicians or general practitioners. Therefore, future reviews can include rural outreach research in other global contexts for other focuses, including the difficulty of implementation.

## 5. Conclusions

Despite the small number of included studies, this systematic review shows that family physician-involved outreach to rural and underserved areas can improve various health outcomes, including anxiety, subjective physical function, and diabetic care. The issues with outreach interventions are the duration and continuity of outreach, the active participation of family physicians and patients, and the focus of outreach participants. Therefore, future studies should use longitudinal research designs to assess outreach programs in rural contexts regarding improving health outcomes, which could motivate more family physicians to conduct outreach programs in their communities.

## Figures and Tables

**Figure 1 ijerph-19-17004-f001:**
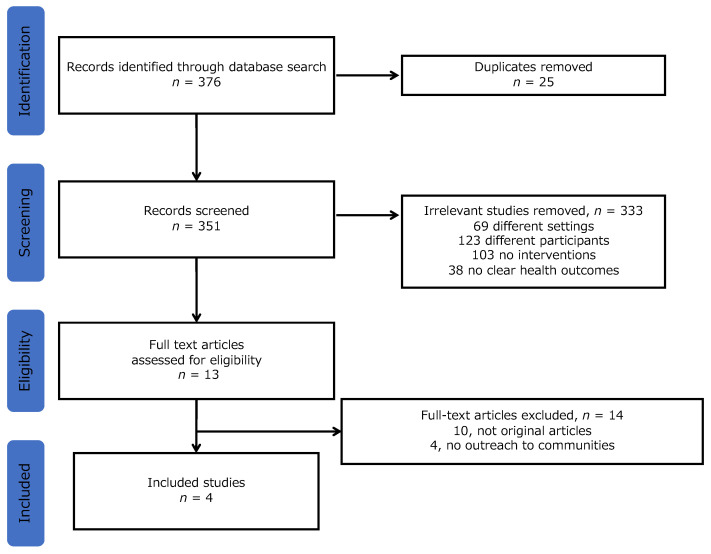
Study selection flow.

**Table 1 ijerph-19-17004-t001:** Inclusion and exclusion criteria.

Criteria	Inclusion	Exclusion
Population	People > 60 to 65 years old	Other people
Setting	Rural or underserved community	Other settings
Types of study	Interventional study	Non-empirical studies (editorial, news, review, conference papers)
Interventions	Outreach including family physicians	Without outreach
Outcome	Health-related	Not health-related
Other	Abstract available Full text available in English	Abstract unavailable Full text unavailable in English

**Table 2 ijerph-19-17004-t002:** Studies included in the review.

Year	Country	Purpose	Study Design	Participants	Health Issues	Interventions	Involved Professionals	Results
2003 [20]	Canada	To assess the effectiveness of a multidisciplinary diabetes outreach service	Pre–post study	Patients living in rural communities	Diabetes control	6-month interventions, home visiting educational message to patients	Family physicians Specialists Nurse educators Dieticians Pharmacists	The intervention was associated with a trend toward 10% improvement in blood pressure.
2010 [21]	Canada	To evaluate the impact of a provider initiated primary care outreach intervention to functional decline.	Randomized controlled trial	Older people in communities	Functional decline	12-month intervention comprehensive initial assessment collaborative care planning health promotion referral to community health and social support services.	Home care nurses Family physicians Patients Family	Changes in functional status and self-rated health did not significantly change.
2018 [22]	USA	To determine the effectiveness of cognitive behavioral therapy to mental conditions	Randomized controlled trial	People in underserved communities	Worry and GAD-related symptom	9-month intervention cognitive behavioral therapy with resource counseling, facilitation of communication with primary care providers about worry/anxiety, integration of religion/spirituality, person-centered skill content and delivery, and nontraditional community providers	General practitioners Healthcare providers Social worker Case manager	Moderate improvements on worry, GAD-related symptoms, anxiety, depression, sleep, trauma-related symptoms, and mental health QOL.
2020 [23]	USA	To show the effectiveness of person-centered wellness home	Randomized controlled trial	People in underserved communities	Physical functions	6-month intervention self-management resource center small group programs plus wellness coaching, as a booster intervention in older adults with chronic diseases.	Family physicians Therapists Community health worker	There was an improvement in self-reported physical functioning, not physical activity.

GAD, general anxiety disorder; QOL, quality of life.

## Data Availability

All relevant datasets used in this study are presented in the manuscript.

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
