# Peer review of "Challenges in Help-Seeking Behaviors among Rural Older People Mitigated through Family Physician-Driven Outreach: A Systematic Review"

_ijerph, 2022, doi:10.3390/ijerph192417004_

Round 1

Reviewer 1 Report

See document uploaded

Author Response

Introduction
Line 31: I think resting must be a typo??

Response:

Thank you for the insightful feedback. We completely agree with the reviewer’s comment and have revised the typo accordingly.

Results
Full references should not be provided in the body of the results section – convention (APA??) should be followed here instead. They should also be numbered. I think there is way too much detail about each of the 4 papers. This could be shortened considerably towards the aim of the paper, the intervention and the outcome/s. Information about the quality of the paper (particularly the rating – I don’t think a need to regularly highlight for example what a 5 means – is not required for each paper – I suggest that an extra column be added to the table that gives the rating for each. And then just summarise the overall of all papers – which I think has been done in one section already. Also any information that is common to all papers (ie they are all RCT in communities) should be summarised at the start of the results (and partly is already I think???) and then not mentioned/repeated for each of the papers in text. The text about each paper needs to be shortened considerably and not repeated.

Response:

Thank you for your insightful feedback. We completely agree with the reviewer’s comment. We deleted the explanations of each research. Additionally, we have enriched the description of the summary as follows:

“3.1. Summary of the Study Results

Considering the study designs, all the studies were comparative interventional studies, and two were randomized controlled trials. The participants were over 60–65 years of age. Two studies were from the United States [22,23] and two were from Canada [20,21]. Considering the health issues, one study dealt with functional decline in usual life [21], one with worry and generalized anxiety disorder [22], one with diabetes control [20], and one with physical dysfunction [23]. The setting of two studies included rural communities [20,21], while the other two studies included underserved areas [22,23]. All the study interventions included multiple professionals: all studies, with family physicians or general practitioners; three, with community workers; one each, with family, patients, social workers, therapists, specialists, dieticians, and pharmacists; and two, with nurses. Considering the outcome measurements, one study measured the QOL based on a questionnaire [22], one measured the worry and general anxiety disorder (GAD) severity with multiple questionnaires [21], one measured the satisfaction of care in diabetes [20], and one measured the perceived and objective physical functions using multiple questionnaires [23]. All of the studies demonstrated unequivocal data considering community outreach; the study grade was rated five.

3.2. Suggested limitations of the interventions

First, the involvement of family physicians in the interdisciplinary teams was limited, considering the implication on care decision-making. A previous study suggested that mutual collaboration among specialists, family physicians, and patients in chronic care can improve the objective health outcomes [20]. Second, the established primary care sufficiently supported older people, and the outreach projects may not improve the health outcomes; thus, outreach projects should be conducted in rural and underserved areas [21]. Third, the involvement of various professionals was limited. A previous study suggested that the involvement of general physicians and other healthcare professionals could improve the health outcomes of the participants [22]. Fourth, the duration of the intervention was short. Continual involvement of family physicians and multiple healthcare professionals in the healthcare of older individuals is needed for objective improvement [23].” (Lines 152–182)

Overall
The wording throughout needs some work – I recognise the authors second language is English but also the structure of sentences/paragraphs is not overly well worded – for example from the discussion:

The need assessment for interventions is essential in public health to establish effective outreach [28]. Therefore, effective outreach should actual needs assessment in communities [28]. Based on this assessment, a comprehensive outreach program should be established.

This could be just one sentence – but even then I am not sure if this is actually coming from the papers they reviewed or from other literature or is just their opinion??? The information is not from the reviewed papers so I am thinking this is opinion – it should link to publications reviewed.

At other times I was not sure what the authors were trying to say – for example:

The duration of outreach interventions may be crucial to improving subjective and objective outcomes in rural communities. This review shows that outreach involving various healthcare professionals and patients improves subjective health conditions, including worry, subjective
physical functions, and diabetic care quality [20,22,23]. However, these studies cannot change the objective outcomes, including objective physical function and other chronic disease outcomes. These results can be attributed to the short intervention duration of outreach. In family medicine, continuity of care is crucial for improving patient health [29,30].

This paragraph seemed to confuse intervention length and type of condition and then brought in continuity of care as being important – this seemed to be a 3rd variable but not coming from the reviewed papers/evidence. I think the discussion needs a comprehensive rewrite.

Also, at times there was a lack of pointing/referencing of the reviewed papers – for example

Here, family physicians were members of outreach interdisciplinary teams. However, the primary interventions were conducted by individuals in other specialties and healthcare professionals, including geriatricians, home care nurses, care workers, and counselors. Each profes-ional can approach a specific set of older people and patients; however, systemic and holistic approaches in collaboration with patients, community members, and family physicians lack in previous studies of this review.

No references are illustrated against different professions from the papers. This should be undertaken throughout to point the reader to what evidence came from what reviewed paper.

Overall I think that there needs to be a rewrite of both results and discussion sections to be more focused – the discussion particularly improving wording and also focusing more on what papers evidence is coming from.

Response:

Thank you for your insightful feedback. We completely agree with the reviewer’s comment. We have revised the contents of the results and discussions sections comprehensively based on the reviewer’s suggestions. In addition, our manuscript was comprehensively reviewed and revised by an English editing company.

Reviewer 2 Report

Congratulations on this good paper, on an important topic.

Minor corrections:

- Include in abstract, information on whether search international (or limited to particular countries)

- Include in method, that no exclusion based upon country

- Include in method time frame of search (April 2000- October 2022)

- Include in method definition of "older"

- Include detail about screening process in method (e.g.  did one person do all title screenings, then a second person check a % of them?)

- List as limitation, not including the word 'regional' in search terms (in some rural settings e.g. Australia that is widely used instead of rural)

Author Response

Congratulations on this good paper, on an important topic.

Minor corrections:

- Include in abstract, information on whether search international (or limited to particular countries)

Response:

Thank you for your insightful feedback. We completely agree with the reviewer’s comment. We have added the relevant content to the abstract.

Lines 15–17

We searched three databases (PubMed, EMBASE, and Web of Science) for international and original interventional articles regarding family physicians involved in outreach to older people in rural or underserved areas from April 2000 to October 2022.

- Include in method, that no exclusion based upon country

Response:

Thank you for your insightful feedback. We completely agree with the reviewer’s comment. We have added the relevant content to the methods section.

Lines 89–91

Original interventional articles were included in international context, whereas conference presentations, reviews, and duplicate articles in the search results were excluded.

- Include in method time frame of search (April 2000- October 2022)

Response:

Thank you for your insightful feedback. We completely agree with the reviewer’s comment. We have added the relevant content to the methods section.

Lines 82–84

In addition, we searched for interventions for HSBs related to family medicine in PubMed, Web of Science, and Embase from April 2000 to October 2022.

- Include in method definition of "older"

Response:

Thank you for your insightful feedback. We completely agree with the reviewer’s comment. We have added the relevant content to the methods section of Table 1.

- Include detail about screening process in method (e.g. did one person do all title screenings, then a second person check a % of them?)

Response:

Thank you for your insightful feedback. We completely agree with the reviewer’s comment. We have added the relevant content to the methods section.

Lines 100–104

Concretely, one of the investigators (R.O.) extracted the data from each original study using a purpose-designed data-extraction form. Two other investigators (T.K. and C.S.) examined the extracted data, which were categorized as follows: country, publication year, participants, purpose, research methodology, health issues, types of intervention, involved professionals, and outcomes concerning outreach.

- List as limitation, not including the word 'regional' in search terms (in some rural settings e.g. Australia that is widely used instead of rural)

Response:

Thank you for your insightful feedback. We completely agree with the reviewer’s comment. We have added the relevant content to the limitations section.

Round 2

Reviewer 1 Report

Congratulations on making the changes - I hope that my suggestions have led to a better manuscript.